# OSTAR: Optimized Statistical Text-classifier with Adversarial Resistance

**Yuhan Yao**[1,2], **Feifei Kou**[1,2*], **Lei Shi**[3], **Xiao Yang**[1], **Zhongbao Zhang**[1], **Suguo Zhu**[4]
**Jiwei Zhang**[1], **Lirong Qiu**[1,2], **Haisheng Li**[5]
[1] School of Computer Science (National Pilot School of Software Engineering), BUPT
[2] Key Laboratory of Trustworthy Distributed Computing and Service, BUPT, Ministry of Education
[3]State Key Laboratory of Media Convergence and Communication, CUC
[4]College of Computer Science and Technology, HDU
[5]School of Computer and Artificial Intelligence, BTBU
*Correspondence: `koufeifei000@bupt.edu.cn`

## Abstract

The advancements in generative models and the real-world attack of machine-generated text(MGT) create a demand for more robust detection methods. The existing MGT detection methods for adversarial environments primarily consist of manually designed statistical-based methods and fine-tuned classifier-based approaches. Statistical-based methods extract intrinsic features but suffer from rigid decision boundaries vulnerable to adaptive attacks, while fine-tuned classifiers achieve outstanding performance at the cost of overfitting to superficial textual feature. We argue that the key to detection in current adversarial environments lies in how to extract intrinsic invariant features and ensure that the classifier possesses dynamic adaptability. In that case, we propose **OSTAR**, a novel MGT detection framework designed for adversarial environments which composed of a statistical enhanced classifier and a Multi-Faceted Contrastive Learning(MFCL). In the classifier aspect, our Multi-Dimensional Statistical Profiling (MDSP) module extracts intrinsic difference between human and machine texts, complementing classifiers with useful stable features. In the model optimization aspect, the MFCL strategy enhances robustness by contrasting feature variations before and after text attacks, jointly optimizing statistical feature mapping and baseline pre-trained models. Experimental results on three public datasets under various adversarial scenarios demonstrate that our framework outperforms existing MGT detection methods, achieving state-of-the-art performance and robust against attacks.The code is available at `https://github.com/BUPT-SN/OSTAR`.

## 1 Introduction

With the remarkable and rapid progress in Large Language Models (LLMs) [1, 2], the quality of machine-generated text has gradually achieved a level that is increasingly comparable to human-authored content. However, the widespread proliferation of such text substantially risks amplifying the dissemination of unreliable or misleading information and diminishing [3, 4, 5] the creative motivation of human authors. As clearly shown in Figure 1(a), which illustrates a real-world detection scenario, MGT texts are often attacked to evade detection, critically challenging the robustness of existing detection methods [6, 7, 8].Therefore, developing reliable detection methods that can robustly distinguish MGT from human-authored content has become a crucial task in societal research.

As shown in Figure 1(b1) and Figure 1(b2), the current MGT detection methods for adversarial environments can be categorized into two approaches: statistical-based methods and classifier-based

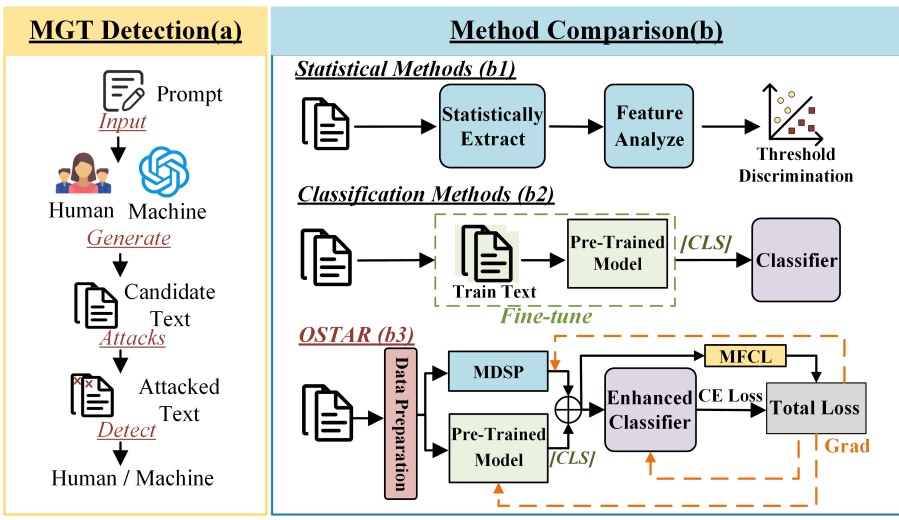

Figure 1: The real-world MGT detection task explanation and the difference between OSTAR and previous detection methods.

methods. While statistical-based methods[9, 10, 11] demonstrate strong zero-shot generalizability across LLMs through intrinsic features (e.g., n-gram distributions, syntactic anomalies), their detection performance is constrained by predefined static thresholds that fail to dynamically adapt to distribution shifts in textual feature patterns under adversarial attacks like lexical substitution or style transfer, leading to significant accuracy degradation. The classifier-based methods[12, 13] rely on pre-trained models (e.g., BERT[14]) to extract deep semantic features through loss function optimization. Although achieving high accuracy on specific datasets, the learned feature representations carry overfitting risks: models tend to capture superficial correlation patterns in training data (e.g., domain-specific sentence structures) rather than the intrinsic differences between human and machine-generated texts. This results in unstable feature representations, leading to a sharp decline in detection performance when distributional discrepancies exist between testing and training data. Collectively, these limitations highlight critical gaps in existing detection paradigms: statistical methods lack adaptation to attacks, classifier-based approaches struggle to capture the intrinsic invariant feature differences between human and machine texts under adversarial attacks.

Given the limitations of current methods in addressing textual attack, single-category detection methods now struggle to achieve high-precision MGT detection in adversarial environments. Our observations reveal that classifier-based methods often demonstrate superior performance on individual datasets, yet exhibit significant performance degradation when confronted with attacked texts. In contrast, although statistics-based detection methods show relatively weaker baseline metrics, certain textual feature statistics (such as Lexical Diversity and Readability) maintain relatively stable deviations between attacked and original texts. This characteristic could serve as an effective mechanism to enhance the robustness of classifier-based methods when operating in adversarial environments. The classifier-based methods can compensate for the shortcomings of statistical methods in feature dynamic adaptation caused by rigid thresholds, as they are capable of developing classification abilities tailored to specific datasets through loss function optimization.

Based on the respective limitations and potential complementarity of the statistical-based methods and classifier-based methods, this paper proposes the **O**ptimized **S**tatistical **T**ext-classifier with **A**dversarial **R**esistance (OSTAR) framework shown in Figure 1(b3). To improve the stability of detection, we design the Statistically Enhanced Classifier, which captures the intrinsic text statistical features thereby enhancing classifier performance. Specifically, we manually design Multi-Dimensional Statistics Profiling(MDSP) to extract and analyze statistical features of texts. These analyzed statistical features are then concatenated with the CLS embeddings generated by pre-trained models to form an enhanced classifier. To enable dynamic feature adaptation in adversarial environments, we categorize attacks into Perturbation and Paraphrases based on whether machine rewriting is involved, distinguishing them by their impact on text characteristics. Guided by this perspective, we design Multi-Feature Contrastive Learning (MFCL). This categorization effectively differentiates attack types by their intrinsic properties, and MFCL significantly enhances the robustness of our method in

adversarial scenarios. Extensive experiments across several datasets and adversarial environments consistently demonstrate that our proposed method outperforms previous approaches, establishing new state-of-the-art performance.

The main contributions of our work are as follows:

- We propose a novel MGT detection framework, OSTAR, which captures the intrinsic differences between machine-generated text and human-authored text and enables dynamic adaptation of detection. To our knowledge, this is the first work to utilize statistical features to guide classifier-based method, enhancing its robustness in adversarial environments.

- We manually designed the MDSP for intrinsic textual characteristics, which effectively captures intrinsic, relatively stable textual multi-dimensional statistical features across diverse scenarios. This mechanism provides a solid foundation for classifier-based methods.

- We categorize attacks into perturbations and paraphrases based on the intrinsic characteristics of their impacts, and design MFCL to comprehensively capture the manifold adversarial effects on text from multiple perspectives, significantly enhancing the overall robustness of our method.

- Through extensive evaluations on three public datasets under diverse adversarial scenarios, OSTAR achieves superior performance and robustness compared to state-of-the-art MGT detection methods.

## 2 Related Work

### 2.1 Machine-generated Text Detection

With the rapid development of LLMs, machine-generated texts, also known as AI-generated texts or Deepfake texts, have made it increasingly difficult for people to distinguish them from human-authored texts [15]. Nowadays, common detection methods can be categorized into three types:watermark-based [16], statistics-based and classifier-based. The watermark-based methods achieve detection of machine-generated text by embedding subtle watermark during the text generation phase and subsequently detecting the presence of watermarks in the generated content [17, 18, 19, 20, 21, 22]. Statistics-based methods often focus on identifying the inherent characteristic differences between human-authored and machine-generated texts by establishing thresholds for differentiation, demonstrating applicability across various models[10, 23, 24, 25, 11]. For example, Eduard[25] proposed distinguishing human and machine texts by analyzing differences in the intrinsic dimensionality of their generated embeddings, achieving stable detection performance across multiple cross-domain and cross-model scenarios. Classifier-based methods can be viewed as a binary classification task, where the detector is typically trained on datasets generated by the target LLM to achieve high-performance detection for the target generative model[26, 27, 28, 29, 30]. For example, SimpleAI[29] fine-tunes RoBERTa, removes biased training data for better generalization, and adds sentence-level data to capture local features. Current MGT detection methods excel on clean data but falter against adversarial attacks. Howerver, many studies[31, 8, 32, 33, 6, 34] have indicated that current MGT detectors exhibit vulnerabilities in real-world detection scenarios, facing challenges to their robustness when subjected to various attacks and adversarial paraphrasing from other pre-trained models. Therefore, maintaining detection robustness in adversarial environments remains a major challenge in current MGT detection tasks.

### 2.2 Contrastive Learning

Currently, in the field of natural language processing, contrastive learning methods can effectively enhance the robustness of frameworks against adversarial attacks[35, 36, 37, 38, 30, 39, 27, 40]. For example, PairCFR[37] enhances the generalization performance of natural language processing tasks when handling counterfactually augmented data by promoting global feature alignment through contrastive learning; DeTeCtive[38] boosts MGT detection generalization through multi-level contrastive learning that identifies cross-sample author style gaps under out-of-distribution task; CoCo[30] tackles sparse training data via contrastive learning, achieving superior performance with minimal training datasets; PESCO[39] addresses the cold-start problem in Zero-Shot text classification by dynamically optimizing document-label matching through a self-training loop of contrastive learning.

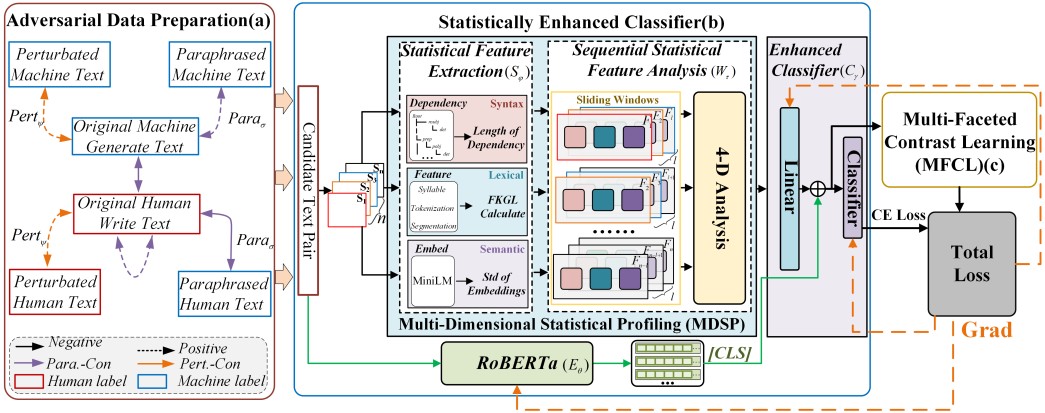

Figure 2: An overview of our OSTAR framework. (a) Adversarial data generation and contrastive learning pair construction (b) Enhancing classifier performance via MDSP (c) Performing contrastive learning and computing loss through MFCL

Existing research demonstrates that contrastive learning exhibits broad applicability and effectively enhances model performance in natural language processing, particularly for MGT classification.

# 3 OSTAR: Framework and Algorithms

We start this section by giving an overview the framework of OSTAR. Then, we detail the specific methods for each step in Sections 3.1 to 3.3. Finally, we will summarize the entire training process into an algorithmic procedure in Section 3.4.

**Abstract Methodology**    As shown in Figure 2, during the training stage, our OSTAR consists of three parts as follows:

- **Part a (Adversarial Data Preparation):** Prior to training, we pre-process the original dataset $O$ to construct the contrastive pairs. Specifically, we apply Perturbation Source $Pert_\psi$ to generate perturbation pairs and Paraphraser $Para_\sigma$ to produce paraphrase pairs.

- **Part b (Statistically Enhanced Classifier):** During the initial stage of the training phase, statistical feature extraction performed on each sentence using Statistical Feature Extraction $S_\varphi$. Once the $S_\varphi$ has processed the entire text, Sequential Statistical Feature Analysis $W_\tau$ will analyze the intrinsic statistical feature with a sliding windows of length $l$ and the 4-D Analysis method. Then, the outputs of $W_\tau$ are projected and concatenated with the $CLS$ token from Pre-Trained Model (RoBERTa is used in this paper) $E_\theta$ for Enhanced Classifier $C_\gamma$ to classify.

- **Part c (Multi-Faceted Contrast Learning):** In MFCL, perturbation contrastive learning and paraphrase triplet contrastive learning are designed for adversarial environments, optimized with updating $\tau$ and $\gamma$.

## 3.1 Adversarial Data Preparation

In terms of adversarial data preparation, as shown in Figure 2(a), we categorize training texts into two types: original human-authored and original machine-generated texts, and texts that may be encountered during detection (divided into Perturbation and Paraphrase). This design is based on the rationale that paraphrases often lead to changes in text ownership, such as transforming text from being generated by LLM to being written by $Para_\sigma$, or converting human-written text into machine-generated content. In contrast, perturbations do not involve ownership alteration but can affect text recognition accuracy. Therefore, we preprocessed the training set by $Pert_\psi$ and $Para_\sigma$, then assigning positive and negative sample pairs according to Figure 2(a), and dynamically constructing the dataset in each epoch.

## 3.2 Statistically Enhanced Classifier

The Statistically Enhanced Classifier is composed of Multi-Dimensional Statistical Profiling in section 3.2.1 and the Enhanced Classifier with a Pre-Trained Model in section 3.2.2.

### 3.2.1 Multi-Dimensional Statistical Profiling (MDSP)

As shown in Figure 2(b), MDSP consists of two components: Statistical Feature Extraction and Sequential Statistical Feature Analysis.

**Statistical Feature Extraction($S_\varphi$)** With the increasingly powerful generative capabilities, relying solely on surface-level statistical features of text has become insufficient for robust MGT detection. While machine-generated texts exhibit certain statistical similarity with human-authored texts in surface patterns, they still demonstrate discernible differences in intrinsic statistical characteristics such as variations in sentence complexity, lexical diversity shifts, and semantic coherence dynamics within a document.

In this paper, we focus on extracting statistical features from three dimensions—syntactic, lexical, and semantic—by quantifying metrics such as the maximum dependency distance per sentence (syntax), the Flesch-Kincaid Grade Level (lexical complexity), and the variance across dimensions in semantic embedding projections (semantic coherence), thereby capturing multi-level discriminative patterns between human-authored and machine-generated texts, the three-dimensional features extraction can be formally formulated as follows:

$$\Phi(s) = \left[ \underbrace{\max_{t \in s} \text{head}(t)}_{\text{Syntax}}, \underbrace{\mathcal{F}_{\text{FK}}(s)}_{\text{Lexical}}, \underbrace{1 - \frac{1}{n-1} \sum_{i=1}^{n-1} \cos(\mathbf{e}_i, \mathbf{e}_{i+1})}_{\text{Semantic}} \right] \tag{1}$$

where, $s$ represents a single sentence in an article, $t$ denotes the tokens in sentence $s$, $\text{head}(t)$ is the head index of token $t$, $\max \text{head}(\cdot)$ represents Maximum dependency distance in parse tree, $\mathcal{F}_{\text{FK}}()$ is the Standard Flesch-Kincaid Grade Level formula, $\{\mathbf{e}_1, ..., \mathbf{e}_n\}$ denotes the sentence embeddings from MiniLM, $\cos(\cdot)$ is inter-sentence cosine similarities.

**Sequential Statistical Feature Analysis($W_\tau$)** When performing statistical feature analysis, we employ a sliding window of length $l$ with a step size $t$ to segment the text. For each feature dimension $d \in \{\text{Syntax}, \text{Lexical}, \text{Semantic}\}$, we design a 4-D Analysis module to analyze the statical feature and use a 4D-feature vector $\mathbf{F}_d \in \mathbb{R}^4$ to store the outputs, aggregating four key measures: mean value ($\mu_d$), standard deviation ($\sigma_d$), autocorrelation coefficient ($\rho_d$), and range ($R_d$). The process is formally formulated as follows:

$$\mathbf{F}_d = [\mu_d, \sigma_d, \rho_d, R_d] = \frac{1}{N} \sum_{k=1}^{N} \begin{cases} \mu_d^{(k)} = \frac{1}{|W^k|} \sum_{y \in W^k} y \\ \sigma_d^{(k)} = \sqrt{\frac{1}{|W^k|-1} \sum_{y \in W^k} (y - \mu_d^{(k)})^2} \\ \rho_d^{(k)} = \text{autocorr}(W^k) \\ R_d^{(k)} = \max(W^k) - \min(W^k) \end{cases} \tag{2}$$

where $W^k$ denotes the sequential statistical features of $k$-th sliding window, $y$ represents the statistical feature of a sentence within the window $W^k$.

For an input text sequence $S = \{s_1, s_2, ..., s_n\}$ with $n$ sentences, we compute window-based statistics across three linguistic dimensions (syntax, lexical, semantic) as follows:

$$\mathbf{T}_S^{\text{raw}} = \bigoplus_{d \in \{Syn, Lex, Sem\}} \left( \frac{1}{|\mathcal{W}_{(d)}|} \sum_{W^k \in \mathcal{W}_{(d)}} \Phi(W^k) \right) \in \mathbb{R}^{12} \tag{3}$$

where, $\mathbf{T}_S^{\text{raw}} \in \mathbb{R}^{12}$ denotes Raw temporal feature vector formed by concatenating window statistics across three dimensions, $\phi(W^k)$ denotes the $k$-th sliding window for $\mu^{(k)}, \sigma^{(k)}, \rho^{(k)}, R^{(k)}$, $\mathcal{W}_{(d)} = \{W_{(d)}^1, \ldots, W_{(d)}^m\}$ represents the Sliding windows for dimension $d$.

**Algorithm 1** OSTAR train process

---

**Require:** Original dataset $\mathcal{O} = (\mathcal{H}, \mathcal{M})$, Perturbation Source $Pert_\psi$, Paraphraser $Para_\sigma$
**Ensure:** Trained parameters $\tau$ (Analyzer), $\gamma$ (Classifier), $\theta$(Pre-Trained Model)

1: **Frozen Components:**
2: $S_\varphi, Para_\sigma, Pert_\psi$
3: **Trainable:** $W_\tau, C_\gamma, E_\theta$
4: **Adversarial Data Preparation:**
5: Generate $\tilde{\mathcal{O}}_{Pert} \leftarrow Pert_\psi(\mathcal{O})$                                   ▷ Perturbation pairs
6: Generate $\tilde{\mathcal{O}}_{Para} \leftarrow Para_\sigma(\mathcal{O})$                                   ▷ Paraphrase pairs
7: Build $\mathcal{D} \leftarrow \mathcal{O} \cup \tilde{\mathcal{O}}_R \cup \tilde{\mathcal{O}}_A$
8: **for** epoch $= 1$ **to** $N$ **do**
9:     **for** batch $(x, \tilde{x}_{Pert}, \tilde{x}_{Para}) \sim \mathcal{D}$ **do**
10:         **Statistical Feature Extraction:**
11:         **for** each sentence $s_i$ in $x$ **do**
12:             Extract $\Phi(s_i) = [\max \text{dep}(s_i), \mathcal{F}_{FK}(s_i), 1 - \frac{1}{n-1}\sum \cos(e_i, e_{i+1})]$
13:         **end for**
14:         **Sequential Statistical Feature Analysis:**
15:         Apply sliding window ($l = 3$, step=1) on $\Phi$ sequence
16:         Compute window 4-D stats: $\mathbf{T}_x^{\text{raw}} = \frac{1}{m}\sum_{k=1}^{m}[\mu_k, \sigma_k, \rho_k, R_k]$
17:         Project: $\mathbf{T}_x^{\text{proj}} = \tanh(W_\tau \mathbf{T}_x^{\text{raw}} + b_t)$
18:         **Statistical Enhance:**
19:         $h_x = E_\theta(x)^{\text{[CLS]}}$                                                        ▷ RoBERTa
20:         $f_x = \text{concat}(h_x, \mathbf{T}_x^{\text{proj}})$                                     ▷ 832-dim
21:         **Multi-Faceted Learning:**
22:         Compute Para-contrast loss: $\mathcal{L}_{\text{Para}} \leftarrow \log \frac{e^{sp/\tau}}{e^{sp/\tau} + \sum_n e^{sn/\tau}}$
23:         Compute Pert-contrast loss: $\mathcal{L}_{\text{Pert}} \leftarrow \beta_a r S_a$
24:         **Parameter Update:**
25:         $\tau, \gamma, \theta - \eta\nabla((1-\lambda) \cdot \mathcal{L}_{\text{CE}} + (\lambda_1 \cdot \mathcal{L}_{\text{Para}} + \lambda_2 \cdot \mathcal{L}_{\text{Pert}}))$
26:     **end for**
27: **end for**
28: **return** $\tau^*, \gamma^*, \theta^*$

---

### 3.2.2 Enhanced Classifier($C_\gamma$)

The output $\mathbf{T}_S^{\text{raw}}$ of Multi-Dimensional Statistical Profiling is used to enhance classifier-based method by projection, the formula is as follows:

$$\mathbf{T}_S^{\text{proj}} = \tanh(\mathbf{W}_t \cdot \mathbf{T}_S^{\text{raw}} + \mathbf{b}_t) \in \mathbb{R}^{64} \quad \text{where } \mathbf{W}_t \in \mathbb{R}^{64 \times 12} \tag{4}$$

where, $\mathbf{T}_S^{\text{proj}} \in \mathbb{R}^{64}$ represents the projected features after nonlinear transformation, $\mathbf{W}_t \in \mathbb{R}^{64 \times 12}$ denotes the learnable projection matrix mapping raw features to latent space, $\tanh(\cdot)$ is the hyperbolic tangent activation function constraining values to [-1, 1].

Then the $E_\theta$ encodes the input text $S$, and the resultant 768-dimensional $CLS$ embedding from the last layer is concatenated with the projected statistical features for classification.

### 3.3 Multi-Faceted Contrast Learning(MFCL)

MFCL can be divided to Paraphrase Contrastive Learning(Para-contrast) and Perturbation Contrastive Learning(Pert-contrast). In Para-contrast, anchors are defined as $H$(original human write text) and $M$(original machine generate text), where the positive and negative samples vary based on the anchor type: when the anchor is $H$, positive samples consist of other original human texts while negative samples are $\tilde{H}_{para}$, whereas for machine-generated anchors ($M$), positive samples are $\tilde{M}_{para}$ and negative samples include any human $H$. In Pert-contrast, only positive samples—corresponding to adversarially attacked versions of the specified text ($\tilde{H}_{pert}$ or $\tilde{M}_{pert}$) are utilized. The Multi-Faceted

Contrast can be formulated as follows:

$$\mathcal{L}_{\text{MFCL}} = \lambda_1 \cdot \underbrace{\sum_{i=1}^{M} \sum_{p \in \mathcal{P}(i)} \log \frac{e^{S_{ip}/\tau}}{\sum_{p' \in \mathcal{P}(i)} e^{S_{ip'}/\tau} + \sum_{n \in \mathcal{N}(i)} e^{S_{in}/\tau}}}_{\mathcal{L}_{\text{Para}}} + \lambda_2 \cdot \underbrace{\sum_{a \in \mathcal{A}(i)} \beta_{ia} r S_{ia}}_{\mathcal{L}_{\text{Pert}}} \quad (5)$$

where, $p \in \mathcal{P}(i)$ and $n \in \mathcal{N}(i)$ denote the positive and negative sample sets for anchor $i$, $S_{ip}$ and $S_{in}$ are similarity scores between anchor $i$ and its positive/negative samples, scaled by temperature $\tau$ to sharpen or soften the contrastive probability distribution; $\mathcal{A}(i)$ is adversarially perturbed sample sets, $S_{ia}$ denotes the is the similarity scores between anchor and perturbed samples, weighted by attack impact ratio $\beta_{ia}$ and scaled by regularization coefficient $r$, $\lambda_1$ and $\lambda_2$ denote the weighting coefficients for $\mathcal{L}_{\text{Para}}$ (paraphrase loss) and $\mathcal{L}_{\text{Pert}}$ (perturbation loss), respectively.

The total loss function is composed of the Cross-Entropy $\mathcal{L}_{\text{CE}}$ loss and $\mathcal{L}_{\text{MFCL}}$, formulated as:

$$\mathcal{L}_{\text{total}} = (1 - \lambda) \cdot \mathcal{L}_{\text{CE}} + \lambda \cdot \mathcal{L}_{\text{MFCL}} \quad (6)$$

where $\lambda$ are weighting coefficients.

### 3.4 OSTAR Algorithm

The overall training process of OSTAR is summarized in Algorithm 1. For a given text, OSTAR extracts its statistical features as supplementary information and concatenates them with embeddings generated by a pre-trained model. These representations are then aligned with Multi-Faceted Contrastive pairs to enhance the model's robustness against attacks.

## 4 Experiments

### 4.1 Experiment Setup

**Datasets and Real-world Attacks**   In this study, we employed three widely-used and moderately challenging datasets: CheckGPT[41], HC3[29], and a cross-domain dataset generated by GLM-130B from the DeepFake[33]. CheckGPT comprises 900,000 multi-domain samples (e.g.news, reviews and articles) generated by ChatGPT[1] using diverse prompts. HC3 is composed of question-answer pairs, where each question includes at least one human-written response and one machine-generated response, focusing on open-ended questions across domains such as finance and medicine. DeepFake contains cross-domain texts generated by various LLMs. To capture real-world MGT diversity, we employ GLM-130B[42] as the representative dataset. Adversarial attacks are categorized into perturbation and paraphrase attacks. For perturbation, the specific 9 perturbations adopted are as follows. For the paraphrase, we employ DIPPER[6] to modify the text. More detailed dataset construction is shown in appendix C.

- **Character-level Perturbations**:
    - Space Insertion: Introduce extraneous whitespace within words (e.g., "hel␣lo")
    - Punctuation Removal: Delete commas, periods, etc. (e.g., "Hello, world!" → "Hello world")
    - Initial Character Case Alteration: Randomize capitalization of word-initial letters (e.g., "apple" → "Apple")
    - Word Merging: Concatenate adjacent words (e.g., "new␣york" → "newyork")
- **Word-level Perturbations**:
    - Keyboard Typos: Simulate typographical errors via adjacent key substitutions (e.g., "house" → "hjuse")
    - Character WordCase: Randomly alter the case of letters within words (e.g., "example" → "ExAmPlE")
    - Spelling Errors: Insert phonetically plausible misspellings (e.g., "because" → "becuz")
    - Adverb Insertion: Add semantically redundant adverbs within sentences (e.g., "He ran quickly" → "He ran *extremely* quickly")

Table 1: Detection performance comparison under original datasets. The best-performing data under each metric has been bolded. Due to dataset balancing, the values of accuracy, recall, and F1 score will be relatively close when the model is well-trained and the architecture is stable.

| Methods | DeepFake | | | CheckGPT | | | HC3 | | |
|---|---|---|---|---|---|---|---|---|---|
| | ACC | Recall | F1 | ACC | Recall | F1 | ACC | Recall | F1 |
| GPT-2 | 87.29 | 90.58 | 88.04 | 81.92 | 83.01 | 80.74 | 90.86 | 90.75 | 89.41 |
| RoBERTa | 91.68 | 91.57 | 91.66 | 88.77 | 87.82 | 88.78 | 94.32 | 94.31 | 94.32 |
| CoCo | 88.03 | 89.59 | 87.58 | 84.55 | 84.90 | 85.97 | 98.42 | 99.31 | 98.50 |
| RADAR | 55.49 | 55.49 | 58.05 | 63.04 | 63.26 | 63.01 | 89.57 | 89.57 | 90.39 |
| Watermark | 86.21 | 90.45 | 88.91 | 75.69 | **97.06** | 72.26 | 94.88 | 94.75 | 95.13 |
| Binoculars | 78.22 | 82.41 | 76.39 | 86.90 | 89.74 | 87.12 | 92.44 | 95.13 | 91.95 |
| PECOLA | 86.29 | 86.19 | 86.29 | 84.58 | 84.96 | 84.51 | 99.23 | 99.25 | 99.24 |
| **OSTAR** | **91.94** | **92.38** | **92.36** | **90.37** | 90.12 | **90.23** | **99.55** | **99.78** | **99.55** |

  – Adverb Append: Attach an additional adverb at the end of sentences (e.g., "The task is done." → "The task is done. *perfectly*")

- **Sentence-level Perturbations**:

  – Sentence Reversal: Invert word order (e.g., "This is a test." → "Test a is this.")
  – Sentence Repetition: Duplicate clauses/phrases (e.g., "I agree. I agree.")

**Evaluation Metric**    To ensure a comprehensive and systematic evaluation of our work, we adopted widely recognized metrics for binary classification task —Accuracy (ACC), Recall, and F1-score (F1)—to evaluate model performance.

**Comparison Methods**    We compare our method with classifier-based state-of-the-art MGT detectors, including: **GPT-2** [43] and **RoBERTa** [44] fine-tuned as binary classifiers (124M/110M parameters); **CoCo** [30] employing contrastive learning with coherence graphs; **RADAR** [26] using adversarial paraphrases for robustness; **Binoculars** [11] leveraging cross-model probability divergence for zero-shot detection; the watermark-based method **Watermark** [20] for reference. Statistic-based methods are excluded due to limited adversarial robustness; **PECOLA** [22] enhancing robustness via core-term perturbation.

To ensure experimental fairness, we utilized data-augmented datasets for training all classifier-based contrastive methods, with the exception of RADAR (whose code was unavailable, necessitating the use of their open-source model). For non-trainable approaches — including statistical-based methods (e.g., Binoculars) and watermark-based techniques (e.g., Watermark) — no training phase was implemented, as their detection mechanisms rely on pre-defined heuristics rather than learnable parameters. All experiments strictly adhered to the original implementations' default training configurations. For methods requiring specialized data preprocessing pipelines (e.g., CoCo), we faithfully executed their prescribed preprocessing steps as outlined in their respective methodologies.

## 4.2    Performance Evaluation

**Evaluation on Original Datasets**    The detection evaluation on the original dataset is shown in Table 1. OSTAR achieved average Acc, Recall and F1 of 93.95%, 94.09%, and 94.04% across the three datasets, outperforming the fine-tuned RoBERTa baseline by average improvements of 2.81% (ACC), 2.85% (Recall), and 2.97% (F1). Moreover, compared to state-of-the-art methods on the MGT detection task, OSTAR attained at least 1.32%, 1.42%, and 1.05% enhancements in ACC, Recall, and F1, respectively, while achieving the best performance across all three datasets. These results demonstrate that our method outperforms the baseline approaches in average detection quality under non-attack scenarios.

Furthermore, as evidenced in Table 1, the incorporation of statistical feature analysis in our method consistently enhances the detection performance of the baseline RoBERTa model across all three datasets, empirically validating the efficacy of our feature analysis approach. Notably, the DeepFake dataset poses the greatest challenge for all models. We attribute this to the fact that GLM-130B—used to generate DeepFake texts—represents a model not adequately considered by conventional pre-

Table 2: Detection performance of OSTAR and the baselines on datasets with attack. We adopt F1-score as the evaluation metric when facing attack, where we categorize attack into Perturbation(Pert.) and paraphrases(Para.). The best-performing data under each metric has been highlighted in bold.

| Methods | DeepFake | | | CheckGPT | | | HC3 | | |
|---|---|---|---|---|---|---|---|---|---|
| | Ori. | Pert. | Para. | Ori. | Pert. | Para. | Ori. | Attack | Para. |
| GPT-2 | 88.04 | 74.23 | 73.41 | 80.74 | 70.58 | 72.56 | 89.41 | 82.72 | 81.63 |
| RoBERTa | 90.12 | 77.10 | 79.00 | 88.78 | 80.62 | 81.59 | 94.32 | 90.27 | 90.95 |
| CoCo | 87.58 | 69.54 | 76.95 | 85.97 | 70.38 | 74.58 | 98.50 | 90.09 | 90.98 |
| RADAR | 58.05 | 48.54 | 47.11 | 63.01 | 60.21 | 67.42 | 49.78 | 47.52 | 58.47 |
| Watermark | 88.91 | 66.35 | 47.01 | 72.26 | 55.16 | 50.07 | 95.13 | 69.05 | 68.16 |
| Binculars | 76.39 | 45.42 | 51.23 | 87.12 | 52.32 | 54.54 | 91.95 | 72.34 | 78.68 |
| PECOLA | 86.29 | 78.13 | 60.08 | 84.51 | 62.64 | 60.71 | 98.35 | 65.09 | 68.82 |
| **OSTAR** | **92.36** | **81.27** | **81.46** | **90.23** | **84.48** | **86.04** | **99.55** | **95.72** | **97.52** |

Table 3: Results of the ablation study on DeepFake dataset with adversarial attacks . We selected Accuracy and F1-score for evaluating, with the best-performing results highlighted in bold.

| Model | Orginal | | Pert. | | Para. | |
|---|---|---|---|---|---|---|
| | ACC | F1 | ACC | F1 | ACC | F1 |
| OSTAR (Plain) | 90.34 | 90.12 | 81.10 | 77.10 | 82.61 | 79.00 |
| OSTAR (Feature Extract) | 90.72 | 90.58 | 81.67 | 77.08 | 82.97 | 79.27 |
| OSTAR (Feature Extract+Analysis) | **92.17** | **92.87** | 82.51 | 80.25 | 83.88 | 80.17 |
| OSTAR | 91.94 | 92.36 | **84.34** | **81.27** | **84.75** | **81.46** |

trained frameworks, thereby requiring re-adaptation to its unique text generation patterns. Our method achieves significant improvements on GLM-130B, demonstrating that the statistical feature analysis approach we proposed serves as a universal feature analysis method for MGT.

**Evaluation on Attacked Datasets**   The ability of detection methods to maintain robustness across diverse attack types constitutes a fundamental research problem, as this capability directly determines their practical applicability in real-world environments. The changes in the model's F1-score across the three datasets under these two types of attacks serve as the robustness measure for our method. As shown in Table 2, our proposed OSTAR framework achieves state-of-the-art F1-scores across all 9 experimental scenarios, which encompass three diverse public datasets (CheckGPT, HC3, and DeepFake) under three distinct conditions (original, perturbation-attacked, and paraphrase-attacked environments). Under adversarial perturbations, OSTAR exhibits a maximum F1 degradation of only 11.09% on the challenging DeepFake-Perturbation subset, significantly outperforming the statistical method Binoculars, which suffers a 30.97% degradation—this stark contrast underscores the inherent limitations of relying solely on static statistical thresholds for real-world robustness. Moreover, with an average F1 degradation of 6.30% against combined perturbation and paraphrase attacks across all datasets, OSTAR surpasses even the most robust baseline model, RoBERTa (which shows 8.49% degradation), thereby highlighting our framework's superior adversarial resilience. Furthermore, OSTAR achieves an average F1 improvement of 4.43% over the RoBERTa baseline, with a maximum gain of 5.5% observed on the HC3 dataset under adversarial attacks—these results collectively demonstrate substantial robustness enhancement and validate the effectiveness of our approach in practical settings.

### 4.3   Ablation Study

To better validate the necessity of each module in our model, we conducted ablation studies on Deep-Fake dataset. The reason why we chose the Deepfake dataset as it showed the sharpest performance drop under adversarial environments, best demonstrating our method's robustness in adversarial scenarios. The ablation study structure designed as follows:

**OSTAR (Plain)** removes the entire Statistical Feature component, retaining only the RoBERTa part. During model training, it also eliminates MFCL while keeping solely the CE loss.

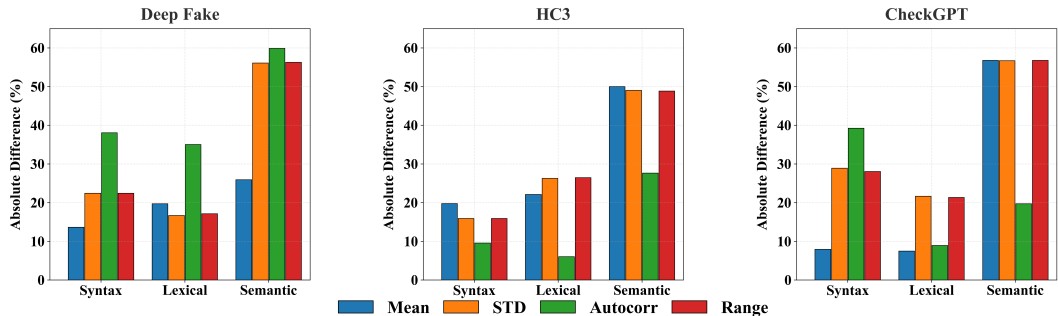

Figure 3: The MDSP performance on three datasets. The vertical axis shows the average absolute difference in MDSP statistical features between human and machine texts across three datasets, demonstrating MDSP's distinct discrimination capability.

**OSTAR (Feature Extract)** retains the three-dimensional feature extraction component but bypasses analysis processing of these features. Instead, it directly applies global average pooling and projects the results to corresponding dimensions for concatenation with RoBERTa's embeddings. The final classification is performed using a linear classification head with CE loss.

**OSTAR (Feature Extract+Analysis)** incorporates the complete Statistical Enhancement component but excludes Multi-Faceted Contrastive Learning during training.

Table 3 shows that both Statistical Enhancement and Multi-Faceted Contrast Learning significantly boost the model's detection performance and adversarial robustness. Our OSTAR achieves optimal performance when facing perturbations and paraphrases attacks. Under these attacks, the ACC drops by 7.60% (perturbation) and 7.19% (paraphrases), while F1-score declines by 11.09% and 10.90% respectively — significantly smaller degradation compared to models without MFCL, demonstrating the necessity of our method. The progressive performance improvements from OSTAR (Plain) → OSTAR (Feature Extract) → OSTAR (Feature Extract+Analysis) validate the effectiveness of MDSP. Notably, using only feature extraction yields marginal gains and even degrades performance under perturbations, likely due to redundant features overlapping with CLS embeddings from pretrained models. While OSTAR with MFCL shows slightly lower performance on the original dataset compared to non-MFCL counterparts (attributed to MFCL training exclusively on attacked data, reducing fidelity to original distributions), this degradation remains within acceptable range (0.23% in acc and 0.51% in F1).

## 4.4 Statistical Feature Evaluation

As demonstrated in Figure 3, our statistical text analysis method, evaluated on three public datasets(HC3, CheckGPT, DeepFake), reveals an average discrepancy of 30.95% between MGT and human-authored texts, proving its significant discriminative power for MGT detection. Extended analysis of our MDSP framework shows in Appendix A. This suite of stable and quantifiable intrinsic statistical features effectively uncovers systematic biases in the linguistic patterns of machine-generated texts. It serves as a critical anchor point for the OSTAR framework, enhancing detection robustness in adversarial environments by compensating for the tendency of pure neural network features to deviate under attacks.

## 5 Conclusion

In this paper, we propose OSTAR, a robust MGT detection framework that synergizes the intrinsic invariant feature extraction capability of statistics-based methods with the dynamic adaptability of classifier-based approaches. Specifically, we design the MDSP module to manually extract and analyze statistical features across multiple intrinsic dimensions, enhancing classification through feature fusion. To address adversarial environments, we categorize attacks into Perturbation and Paraphrase based on their impact mechanisms, and accordingly develop MFCL to improve robustness by disentangling adversarial effects through multi-perspective feature alignment. Extensive experiments across three public datasets with 9 kinds of perturbations and a paraphraser, validate the effectiveness of OSTAR and demonstrate its robustness under various attacks.

## Acknowledgments

This work was supported by the National Natural Science Foundation of China (62002027, 62472042,62572075,62277001), the Beijing Municipal Natural Science Foundation (L257023, L233034), the Fundamental Research Funds for the Central Universities (No. CUC25SG013) the Fundamental Research Funds for the Beijing University of Posts and Telecommunications(Grant No.2025TSQY01), the National Natural Science Foundation of China Youth Project (Grant No. 62402057) and the State Key Laboratory of Cyberspace Security Defense (No.2025-C08).

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

# A  Statistical Feature Evaluation

To further verify that the features extracted using MDSP maintain a certain level of stability compared to the original text after being subjected to perturbations and paraphrases, we employed kernel density estimation (KDE) plots to evaluate each statistical feature.

## A.1  Variations in the Syntax Features Extracted by MDSP

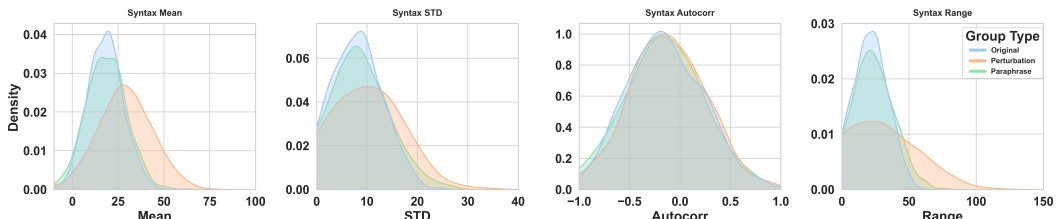

Figure 1: Variations in the Syntax features extracted by MDSP on human texts

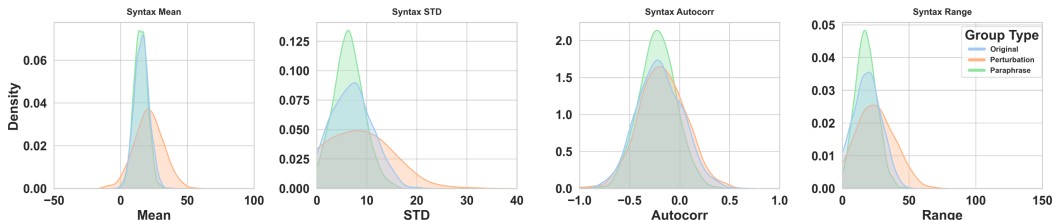

Figure 2: Variations in the Syntax features extracted by MDSP on machine texts

As revealed by the KDE plots, texts subjected to perturbation attacks (e.g., sentence repetition) exhibit substantial divergence from original human texts in the syntactic statistical features extracted by MDSP. This discrepancy likely stems from how repetitive patterns disrupt syntactic regularity, yet they maintain sufficient similarity for discriminative feature learning. In contrast, paraphrased texts demonstrate minimal syntactic deviation from original human-authored content, which may hinder classification accuracy and thus necessitates more sophisticated classifier learning to capture subtle discriminative patterns.

## A.2  Variations in the Lexical Features Extracted by MDSP

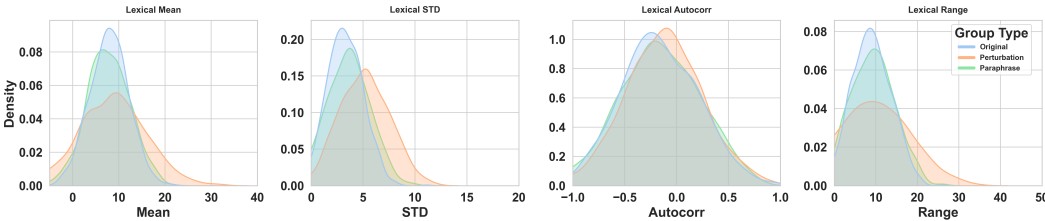

Figure 3: Variations in the Lexical features extracted by MDSP on human texts

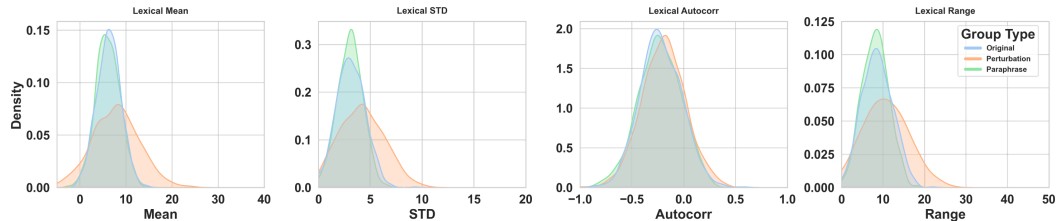

Figure 4: Variations in the Lexical features extracted by MDSP on machine texts

At the lexical level, the three text categories (original, perturbed, and paraphrased) exhibit the closest similarity in auto-corrected (Autocorr) feature representations, which significantly aids in distinguishing perturbation-attacked texts. In contrast, paraphrased texts show minimal divergence across four lexical complexity metrics (e.g., type-token ratio, entropy), yet this subtle variation retains discriminative relevance for identifying machine-generated paraphrased content.

## A.3 Variations in the Semantic Features Extracted by MDSP

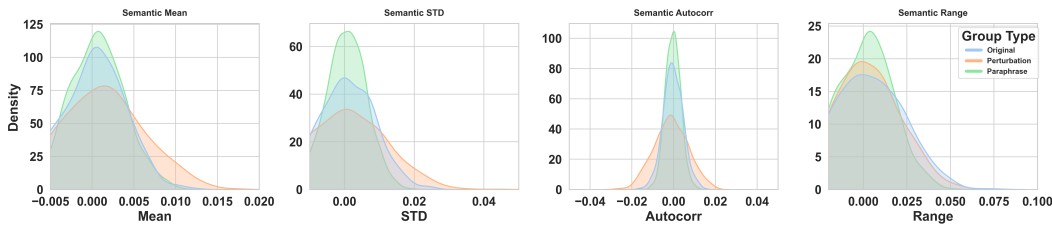

Figure 5: Variations in the Semantic features extracted by MDSP on human texts

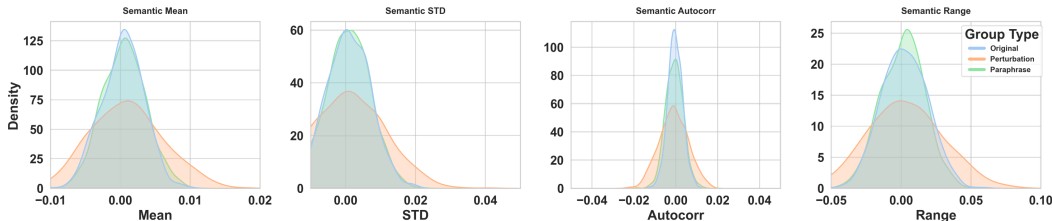

Figure 6: Variations in the Semantic features extracted by MDSP on machine texts

At the semantic level, paraphrased human texts exhibit significant deviations from original human texts across four key metrics (e.g., semantic coherence, entity consistency), which effectively enhances their discriminability. In contrast, paraphrased machine-generated texts maintain close statistical alignment with original machine texts in terms of mean values, standard deviations (STD), and value ranges, thereby enabling robust identification of machine-generated paraphrased content.

## B Limitations

Our method achieves generalized detection against various pre-seen attacks in the training set, but struggles to maintain robustness when confronted with unforeseen attacks such as multiple paraphrases, combined attacks, etc. Another limitation is that the sliding window approach for statistical feature analysis imposes requirements on text length (e.g., short single-sentence passages cannot support effective statistical feature extraction).

## C  Detailed Construction of Dataset

For the three datasets, we constructed both original and adversarially attacked versions. The detailed composition of our original datasets is presented in Table 4, while the configuration of the attacked datasets (generated via perturbation/paraphrases attacks) is summarized in Table 5. The two-tuple (human, machine) in the table represents the number of human texts and the number of AI texts. The symbol "× " in Table 5 indicates that nine distinct adversarial perturbation methods (shown in appendix E) were applied to each original sample, resulting in a tenfold expansion of the dataset size. To mitigate computational overhead caused by this exponential growth, we selected part of each dataset to include 500 human-authored and 500 machine-generated samples for balanced training and testing.

Table 4: Dataset composition during the Evaluation on Original Datasets experiment.

| Dataset | Train | Test | Valid |
|---|---|---|---|
| CheckGPT | (2000, 2000) | (1921, 2078) | (2500, 2500) |
| HC3 | (5000, 5000) | (5000, 5000) | (2000, 2000) |
| DeepFake | (2000, 2000) | (2000, 2000) | (2000, 1000) |

Table 5: Perturbation Methods and Their Intensities

| Attack Name | Intensity | Explanation |
|---|---|---|
| Space Insertion | 5-10 spaces | Inserts 5-10 spaces randomly in text |
| Punctuation Removal | Single char | Removes last punctuation character from text |
| Initial Character Case Alteration | 10% | Randomly alter 10% of word initial characters |
| Word Merging | 20% | Randomly merge 20% of adjacent words |
| Keyboard Typos | 10% | Generates typos in 10% characters using adjacent keys |
| Character WordCase | 20% | Randomly changes case for 20% of words |
| Spelling Errors | 3 | Introduces 3 spelling errors in each sentence |
| Adverb Append | 1 | Appends one adverb to each sentence |
| Sentence Reversal | 10% | Reverses text segments using 3-word pivots in 10% sentence groups |
| Sentence Repetition | 3 | Selects 3 sentences to repeat |

Table 6 shows the detailed composition of the attacked datasets used in the Evaluation on Attacked Datasets experiment, where the "×" symbol indicates the multiplication factor applied to the original sample counts due to adversarial attacks.

Table 6: Dataset composition during the Evaluation on Attacked Datasets experiment.

| Attack Type | Data Split | | |
|---|---|---|---|
| | Train | Test | Valid |
| Perturbation | (500×10,500×10) | (500×10,500×10) | (200×10,200×10) |
| Paraphrase | (500×2,500×2) | (500×2,500×2) | (200×2,200×2) |

## D  Training Details

Our implementation uses RoBERTa as the base pretrained model. Identical attack procedures were applied to both training and test sets. For the MDSP, we set length $l = 3$. The contrastive learning component employs a weight of 0.02 and temperature coefficient of 0.05. Training utilizes the Adam optimizer with learning rate $1 \times 10^{-5}$ and adam epsilon of $1 \times 10^{-8}$. Our model was trained on an NVIDIA RTX 3090 GPU, requiring approximately 17GB of VRAM with a batch size of 4, making it feasible to implement under most laboratory conditions.

