# OpenReview forum: "OSTAR: Optimized Statistical Text-classifier with Adversarial Resistance"
_NeurIPS.cc/2025/Conference — NeurIPS 2025 poster_

### Official Review · Reviewer_8bFh · 2025-06-26

**Clarity:** 4
**Significance:** 4
**Originality:** 4
**Rating:** 5
**Confidence:** 4

**Summary:**

The paper introduces a novel method for detecting machine-generated text in adversarial settings. It combines statistical feature extraction with a neural classifier by integrating a multi-dimensional profiling module and a contrastive learning objective tailored to different types of text attacks. The approach is evaluated on several benchmarks and shows improved robustness and overall performance compared to prior methods.

**Questions:**

The MDSP relies on manually designed features. Could the authors comment on whether other linguistic metrics were considered, or whether data-driven selection could further enhance performance?

**Ethical Concerns:**

["NO or VERY MINOR ethics concerns only"]

**Final Justification:**

The authors provide detailed experiments and statistics which answer my questions. I decide to raise my score to 5.

**Limitations:**

yes

**Quality:**

3

**Strengths And Weaknesses:**

Strengths:
1. The proposed framework introduces a well-motivated combination of statistical profiling (MDSP) and task-specific contrastive learning (MFCL). This hybrid approach is original and effectively addresses the limitations of both traditional statistical detectors and purely classifier-based models under adversarial settings.
2. The experiments are comprehensive. The proposed method achieves state-of-the-art results across multiple datasets.The ablation studies and detailed breakdown of results provide convincing evidence for the effectiveness of each component.
3. The paper is clearly written and well-structured. The methodology is described in sufficient detail, including algorithmic steps, training procedures, and statistical feature definitions, making it easy to follow and likely straightforward to reproduce.

Weaknesses:
1. While the paper evaluates both perturbation and paraphrase-based attacks, the set of perturbation techniques is relatively narrow and mainly consists of syntactic or lexical-level changes. More advanced or semantic-preserving adversarial strategies (e.g., many word-level adversarial attacks) are not considered, which may limit the generalizability of the findings.
2. The statistical features used are manually defined and fixed. It remains unclear whether these choices are optimal or how sensitive the model is to variations or additional feature dimensions.

---

> ### Author Rebuttal · Authors · 2025-07-31
>
> **Weakness 1**:
> Thank you for your observation; when implementing text rewriting, we perform full-document perturbations and have specified in the appendix that our perturbations include sentence-level attacks such as Sentence Reversal: inverting word order and Sentence Repetition: duplicating clauses/phrases—we maintain our adversarial selections and testing demonstrate certain universality.
>
> **Weakness 2**:
> Thank you for your observation; we have adopted multiple word-level attacks here including Word Merging, KeyBoard Typos, Character WordCase, Spelling Errors etc. as shown below, and can additionally incorporate BERT-based adjective replacement attacks if necessary.
> | Attack Name                     | Intensity       | Explanation                                                                 |
> |---------------------------------|-----------------|-----------------------------------------------------------------------------|
> | Space Insertion                 | 5–10 spaces     | Inserts 5–10 spaces randomly into text                                     |
> | Punctuation Removal             | Single character| Removes final punctuation character from text                              |
> | Initial Character Case Alteration| 10%             | Randomly alters casing for 10% of words                                   |
> | Word Merging                    | 20%             | Randomly merges 20% of adjacent words                                     |
> | Keyboard Typos                  | 20%             | Generates typos in 20% characters using adjacent-key substitutions         |
> | Character Word Case             | 20%             | Randomly changes case for 20% of words (e.g., UPPER/lower)                |
> | Spelling Errors                 | 3 per sentence  | Introduces 3 spelling errors in each sentence                              |
> | Adverb Append                   | 1 per sentence  | Appends adverb to the end of each sentence                                |
> | Sentence Reversal               | 10% of groups   | Reverses segments using 3-word pivots in 10% of sentence groups           |
> | Sentence Repetition             | 4 sentences     | Selects and repeats 4 distinct sentences
>
> **Weakness 3**:
>
> Thank you for your feedback! All statistical feature dimensions in our proposed method demonstrate utility, as evidenced by the ablation studies on feature dimensions in the table below. The results reveal that, for these specific features, our designed approach represents the optimal combination identified thus far. Nevertheless, our primary goal is not to establish a definitive framework for constructing statistical features, but rather to propose a unified methodology that integrates statistical feature-based analysis with classifier-based techniques to enhance AI-generated text detection. We hope this work inspires further exploration into more effective statistical modeling approaches, motivating researchers to advance this field by combining statistical and classification-based strategies.
> | 4-D features  | ACC   | Recall | F1   |
> |---------------|-------|--------|-------|
> | u             | 89.92 | 98.25  | 89.05 |
> | μ+σ           | 90.06 | 97.71  | 90.93 |
> | μ+σ+ ρ        | 90.43 | 98.69  | 90.70 |
> | σ+ ρ+ R       | 91.51 | 96.40  | 91.90 |
> | μ+σ+ ρ+ R     | 91.94 | 97.38  | 92.36 |
>
> **Question 1**:
> Thank you for your question;！In related studies we have incorporated alternative textual quality assessment methods as statistical features—such as evaluation metrics spanning literary sophistication and textual sentiment domains—and integrating these into our framework constitutes an effective feature selection strategy where valid features should further enhance performance when added to the system.

---

### Official Review · Reviewer_6FiE · 2025-06-27

**Clarity:** 4
**Significance:** 3
**Originality:** 4
**Rating:** 6
**Confidence:** 5

**Summary:**

This paper proposes the OSTAR framework to enhance robustness in MGT detection within adversarial environments.  By designing statistically enhanced classifier with Multi-Faceted Contrastive Learning , addressing limitations of existing methods. Experiments demonstrate OSTAR's performance and robustness.

**Questions:**

1. Are there any possible solutions or ideas for addressing the issues of multiple attacks and the short text problem mentioned in the limitation section?
2. Can you provide more tables to present the concrete metrics of each attack's intensity and impact, making the attacks more explicit?
3. Can you provide a comparison of the increased training time when using contrastive learning strategies?
4. During training, the authors balanced the quantity of human-generated text and machine-generated text. If the ratio between these two becomes imbalanced, how would that impact the model’s performance?

**Ethical Concerns:**

["NO or VERY MINOR ethics concerns only"]

**Final Justification:**

The authors have addressed all my concerns. This paper presents a novel method with a statistically enhanced classifier and Multi-Faceted Contrastive Learning. I am in favor of acceptance of this paper.

**Limitations:**

yes

**Quality:**

3

**Strengths And Weaknesses:**

Strengths
1. The motivation of this work is clear: addressing the vulnerability of existing MGT detection methods in adversarial environments.
2. Theoretical innovation: categorizing attacks based on whether they affect their originating sources, thereby effectively enhancing the response capability of contrastive learning against attacks.
3. Novel Framework: OSTAR innovatively designs a statistically enhanced classifier and Multi-Faceted Contrastive Learning, resolving limitations of rigid decision boundaries in statistical methods and overfitting in classifier-based approaches under adversarial environments.
4. Comprehensive Experiments: This paper evaluate the performance of OSTAR on three public datasets with seven baselines under nine attack scenarios. Special experiments are used to verify the statistical feature distinguishability and the attack impact of statistical feature.
5. Clear structure: The structure is well organized with well-defined figures and tables, making it easy for readers to follow.

Weaknesses:
1. The limitations section notes the model’s non-robustness against multi-attack scenarios and compromised MDSP extraction capability with short texts, without proposing potential solutions.
2. The text only provides the types of attacks, but the intensity of the attacks is not given, and the description of the adversarial environment is insufficient.
3. Although this contrastive learning can enhance model performance, it substantially increases computational consumption. The paper does not provide relevant data or solutions for this issue.

---

> ### Author Rebuttal · Authors · 2025-07-31
>
> **Question 1**:
> Thank you for your question. Regarding defense against multiple adversarial attacks, a viable solution involves incorporating combinations of various attack methods into the training data during model development to enhance the model's ability to recognize such complex scenarios. As for the short-text detection challenge, we can introduce additional statistical or contextual features independent of text length. These improvements would significantly enhance the model’s performance in complex attack scenarios and short-text situations.
>
> **Question 2**:
>  Thank you for your question! We've detailed adversarial attack metrics in Appendix E. Below is the enhanced specification table:
>
> | Attack Name                     | Intensity       | Explanation                                                                 |
> |---------------------------------|-----------------|-----------------------------------------------------------------------------|
> | Space Insertion                 | 5–10 spaces     | Inserts 5–10 spaces randomly into text                                     |
> | Punctuation Removal             | Single character| Removes final punctuation character from text                              |
> | Initial Character Case Alteration| 10%             | Randomly alters casing for 10% of words                                   |
> | Word Merging                    | 20%             | Randomly merges 20% of adjacent words                                     |
> | Keyboard Typos                  | 20%             | Generates typos in 20% characters using adjacent-key substitutions         |
> | Character Word Case             | 20%             | Randomly changes case for 20% of words (e.g., UPPER/lower)                |
> | Spelling Errors                 | 3 per sentence  | Introduces 3 spelling errors in each sentence                              |
> | Adverb Append                   | 1 per sentence  | Appends adverb to the end of each sentence                                |
> | Sentence Reversal               | 10% of groups   | Reverses segments using 3-word pivots in 10% of sentence groups           |
> | Sentence Repetition             | 4 sentences     | Selects and repeats 4 distinct sentences                                 |
>
> **Question 3**:
>
> ​ Thank you for your question! The contrastive learning strategy does indeed significantly increase training time and hardware resource requirements. Here we provide a comparison between training without contrastive learning and training with it: specifically, VRAM usage increases by approximately 50%, while training time rises by roughly 1.5 times.
>
> | Training Method   | Batch Size | Train Time       | VRAM Usage |
> |-------------------|------------|------------------|------------|
> | No Contrastive    | 16         | 60.2 min/epoch   | 13.7 GB    |
> | ​Contrastive**  | 16         | ​151 min/epoch| ​19.6 GB|
>
> **Question 4**:
>
> Thank you for your question; maintaining balanced proportions between human texts and machine-generated texts during training is crucial as severe data imbalance significantly undermines model performance: excessive machine texts cause the model to overlearn mechanical patterns, generating outputs lacking human linguistic diversity and naturalness while degrading recognition of human text features and lowering human text recall rates; such imbalance distorts learning objectives, creating biases toward dominant data sources and ultimately impairing generalization capabilities and overall performance in text generation, content detection, or hybrid text processing tasks.

---

> > ### Comment · Reviewer_6FiE · 2025-08-01
> >
> > The authors convincingly addressed all four key questions. They proposed practical solutions for multi-attack robustness and short-text challenges, strengthened reproducibility through explicit attack parameter tables, clarified critical data-balance mechanisms with measurable impact, and quantified the trade-offs of contrastive learning. These substantive enhancements significantly elevate the paper’s technical depth and practical utility, therefore I will raise the score to 6.

---

### Official Review · Reviewer_rAmU · 2025-07-03

**Clarity:** 3
**Significance:** 2
**Originality:** 3
**Rating:** 4
**Confidence:** 3

**Summary:**

This paper states that the key of machine-generated text detection lies in how to extract intrinsic invariant features and ensure the dynamic adaptability of the classifier. Thus, it proposes OSTAR to solve the problem. The experiment results shows its effectiveness in three datasets.

**Questions:**

According to the strengths and weaknesses part, I'd like to learn more about how the proposed method solves the intrinsic feature capturing problem.

**Ethical Concerns:**

["NO or VERY MINOR ethics concerns only"]

**Final Justification:**

The author answers the question about "intrinsic properties extraction" and talks about the other issues. Thus, I raise my rating.

**Limitations:**

Yes.

**Quality:**

3

**Strengths And Weaknesses:**

This paper is well organized, and the proposed method is well detailed. The authors evaluate several novel methods on three representative datasets, which provides a comprehensive view of the methods efficacy.

The author states that the key of MGT detection lies in how to extract instrinsic invariant features and ensure the dynamic adaptability of the classifier. It also claims that "classification methods tend to capture superficial correlation patterns rather than the intrinsic differences in training data". I think "capturing superficial correlation patterns in training data" is actually a common property of deep learning methods, and the design of a validation set is exactly to address the problem&mdash;proving that the method are not just overfitting the training data, but also learning some "intrinsic" properties so that it can perform well on the test set.

If the author claim that the proposed method captures the intrinsic patterns of MGT, I would expect to see an explicit design in the proposed framework, or some experimental analysis to support the claim. I think the three dimension features used in the MDSP component (syntactic, lexical, semantic) or the two types of contrast learning in MFCL (para-contrast, pert-contrast) might not be so explicit in solving the "intrinsic" problem, and the existing experiment might not provide a strong provement to support the statement. I think the performance of existing methods is already high enough on the three datasets, so simply improving the metrics further may not bring significant benefits to the community.

Besides, I think Fugure 1 may not by a good teaser figure to introduce the task and the proposed method to the reader. Line 25 claims that "As shown in Figure 1(a), in real-world scenarios, MGT texts are often attacked to evade detection". But the Figure 1(a) literally doesn't show that MGTs are often attacked, it just presents an attack flowchart. And Figure 1(b) lists two old types of pipelines, along with the proposed method. However, it doesn't clearly shows the difference, or the key modification, that distinguish the proposed methods from the others.


Additionally, I think the paper need a proofread to correct typos in the script:

* Table 3 highlights the performance of OSTAR, but OSTAR (Feature Extract+Analysis) has a higher performance.
* Line 25-26 "MGT texts" is duplicated.
* Line 181 The sentence ends with a comma.
* Line 216 "e.g.news, reviews..." is missing a comma.
* Line 224 "Evaluation Metric" -> "Evaluation Metrics"
* Whether there sould be a space before a left parenthesis (in many places) and an em dash (Line 225) should be consistent thoughout the paper.

Moreover, the code is not submitted with the manuscript, instead it will be publicly available upon publication. But the author claims "Yes" in the paper checklist that the paper provides open access to the data and the code. I'm not sure if it's proper.

---

> ### Author Rebuttal · Authors · 2025-07-31
>
> **Answer 1**:
> Our "intrinsic invariant features" refer to the universal characteristics of machine-generated text—specifically, those properties that maintain relative stability even after text undergoes rewriting or perturbation by another text generator. While validation set design partially addresses the issue of "capturing superficial correlative patterns in training data" (since practitioners adjust training methods—such as epochs, learning rates, and regularization—based on validation performance), this remains fundamentally a training strategy. Merely tuning hyperparameters without structurally enhancing the network's capability yields limited improvement. Therefore, we explicitly extracted statistically stable features for identifying AI-generated text and implemented the MFCL contrastive learning strategy to capture the intrinsic relationships between attacked texts and original texts. This approach significantly boosts the model's capacity to recognize and interpret inherent characteristics of AI-generated content.
> Moreover, optimizing training strategies via validation sets and enhancing model performance through architecture improvements represent two distinct pathways for capturing intrinsic task features: one evaluates whether the training process achieves intrinsic feature extraction, while the other expands the model’s feature extraction capability by enriching architecture and feature categories. These methods are mutually non-conflicting and can be applied concurrently. In our training, we indeed leveraged validation sets to optimize model performance.
>
> **Answer 2**:
> In Figure 3 and Appendix A, we provide the average differences between human and machine features under the MDSP method across three datasets, along with the post-attack distributions of both human and machine texts from these datasets. These results demonstrate that our proposed method captures intrinsic patterns of machine-generated text, effectively distinguishing human text from machine text—even when facing various attacks.
> Though statistical feature ranges shift in both human and machine texts after attacks, their distributions remain analogous; simultaneously, distinctions between the two text types persist. Through further training, these patterns become capturable as intrinsic regularities. Therefore, we assert that our original paper provides experimental evidence confirming our method's extraction of intrinsic patterns. We can supplement additional experimental analyses in the appendix to help readers better understand how our method captures inherent distinctions between AI-generated text and human text.
>
> **Answer 3**:
> In fact, current methods perform well on the original datasets but exhibit poor performance on attacked datasets, as shown in the Para. and Pert. columns of Table 2. Here, we present results of existing methods on the CheckGPT dataset from Table 2, which demonstrate significant room for improvement in performance. Therefore, we believe that advancing this metric under adversarial conditions holds meaningful significance for the progress of the field.
> | Methods   | Ori.  | Pert. | Para. |
> |-----------|-------|-------|-------|
> | GPT-2     | 80.74 | 70.58 | 72.56 |
> | RoBERTa   | 88.78 | 80.62 | 81.59 |
> | CoCo      | 85.97 | 70.38 | 74.58 |
> | RADAR     | 63.01 | 60.21 | 67.42 |
> | Watermark | 72.26 | 55.16 | 50.07 |
> | Binculars | 87.12 | 52.32 | 54.54 |
> | PECOLA    | 84.51 | 62.64 | 60.71 |
>
> **Answer 4**:
> Figure 1a provides background on the AI-generated text detection task in adversarial settings. We included this figure to give readers a clearer understanding of our task. The current description of Figure 1a in the main text is not precise enough. Originally, we introduced it as follows:
> “As shown in Figure 1(a), in real-world scenarios, MGT texts are often attacked to evade detection, critically challenging the robustness of existing detection methods[6, 7, 8].”
> We will revise it to:
> “In real-world scenarios, users frequently employ adversarial tactics to evade detection when attempting to bypass machine-generated text detectors. This poses significant challenges to the robustness of existing detection methods [6, 7, 8]. To explain how these adversarial tactics work, which can help clarify our detection task, Figure 1(a) provides an introduction to the process regarding the robust MGT detection task background.”
>
> In Figure 1b, our method and existing methods based on statistical features and classifiers are already connected to some extent (using color) and show differentiation (with clear procedural differences). However, we can add more explicit annotations in the future to highlight our innovation.
>
> **Answer 5**:
> Thank you for your correction. Experimentally, OSTAR indeed performs best on para and pert datasets. However, its slight degradation in performance on the original data—caused by the necessary incorporation of contrastive and adversarial learning—explains the differences shown in the table. We will further revise the manuscript and double-check the content to eliminate any oversights or basic errors.
>
> **Answer 6**:
> We apologize for not providing the code link. Since including any external links is prohibited during the review process, it was not included in the manuscript. The code and relevant links will be made accessible online in the camera-ready version upon acceptance.

---

> > ### Comment · Reviewer_rAmU · 2025-08-07
> > **Reply**
> >
> > Thank the author for the detailed reply. Thank the author for the clarification. I will reconsider my score after the discussion with AC.

---

### Official Review · Reviewer_ryiM · 2025-07-03

**Clarity:** 3
**Significance:** 3
**Originality:** 3
**Rating:** 5
**Confidence:** 3

**Summary:**

The paper introduces OSTAR, a novel framework designed to detect machine-generated text (MGT) with enhanced robustness against adversarial attacks. It introduces Multi-Dimensional Statistical Profiling to capture invariant patterns and Multi-Faceted Contrastive Learning to handle adversarial attacks. The MDSP captures invariant patterns across scenarios, while attack-specific contrastive learning mitigates adversarial effects. Evaluations on three datasets show state-of-the-art performance with enhanced accuracy and robustness.

**Questions:**

1. Which of the 3 MDSP dimensions (syntax/lexical/semantic) contributes most to robustness? Are all 4-D features (μ, σ, ρ, R) necessary?
2. How does OSTAR perform against unseen attack combinations (e.g., perturbation + paraphrase) or sophisticated adversarial attacks beyond the 9 pre-defined perturbations?
3. Is it possible to use LLM for machine-generated text detection and how effective is it?

**Ethical Concerns:**

["NO or VERY MINOR ethics concerns only"]

**Final Justification:**

The author's detailed rebuttal has effectively addressed the key points of concern raised earlier.

**Limitations:**

yes

**Paper Formatting Concerns:**

This paper has no formatting errors.

**Quality:**

3

**Strengths And Weaknesses:**

Strengths
1. The paper combines Multi-Dimensional Statistical Profiling (MDSP) with classifier-based methods, improving robustness in detecting machine-generated text under adversarial attacks.
2. The structure is clear and well-organized, with detailed technical descriptions, pseudocode, that enhance readability and reproducibility.
3. Extensive experiments on three datasets show strong performance, achieving high accuracy and robustness against various adversarial attacks.

Weaknesses
1. The paper lacks an experimental analysis of hyperparameters and a detailed representation of implementation details, such as attack impact ratio, regularization coefficient, etc.
2. The relationship between L_Para and L_Pert needs clearer intuition.
3. While MFCL's categorization is novel, using contrastive learning for robustness itself builds on prior work.

The paper has some minor problems.
- There are many places in the paper where spaces are missing, such as “text(MGT)” and “Learning(MFCL)” in the abstract.
- There are some grammar problems in the paper, such as “an overview the framework of OSTAR” in line 119.

---

> ### Author Rebuttal · Authors · 2025-07-31
>
> **Weakness 1**:
> Our method comprises MDSP feature extraction, statistical features, and the MFCL optimization strategy. Key hyperparameters include MDSP's sliding window size (adjusts feature scope; small sizes limit text-variation reflection, large sizes introduce noise) and MFCL's contrastive learning weight (balances robustness against attacks vs. primary task focus). Experiments validating these parameters are tabulated below.
> Regarding the sliding window size, a window size of 3 achieved optimal balance (accuracy ACC: 91.94%, F1-score: 92.36%), enabling sufficient context feature capture while avoiding interference from distant text noise. Excessively small windows lacked contextual awareness, whereas larger windows (size 5) introduced irrelevant signals and imposed stringent text length requirements, performing inadequately on shorter texts. This sensitivity experiment demonstrated that a window size of 3 achieves the most robust feature extraction by optimizing local context modeling.
>
> | Sliding_windows | ACC   | Recall | F1    |
> |----------------|-------|--------|-------|
> | 1 (no windows) | 90.64 | 97.27  | 91.23 |
> | 2              | 91.29 | 96.19  | 91.79 |
> | 3              | 91.94 | 97.38  | 92.36 |
> | 5              | 91.32 | 98.82  | 90.42 |
>
> Regarding the sensitivity of the contrastive learning weight, this parameter exhibits high sensitivity. Performance improves significantly within the 0.05 to 0.02 range, but declines when the weight falls below 0.02. The final selection of 0.02 is based on the simultaneous peaking of both metrics at this value (Pert=81.27, Para=81.46), representing the optimal balance between the effect of contrastive learning and primary task performance.
>
> | Contrastive Learning Weight | Pert  | Para  |
> |-----------------------------|-------|-------|
> | 0.2                         | 77.81 | 78.29 |
> | 0.05                        | 80.11 | 80.04 |
> | 0.02                        | 81.27 | 81.46 |
> | 0.01                        | 80.92 | 80.93 |
>
> Regarding further implementation details, we provide a table of attack impact rates here; if the paper is accepted for publication, we will include it in the main text to enhance the clarity of implementation details.
>
> | Attack Name                     | Intensity       | Explanation                                                                 |
> |---------------------------------|-----------------|-----------------------------------------------------------------------------|
> | Space Insertion                 | 5–10 spaces     | Inserts 5–10 spaces randomly into text                                |
> | Punctuation Removal             | Single character| Removes final punctuation character from text                              |
> | Initial Character Case Alteration| 10%             | Randomly alters casing for 10% of words                                   |
> | Word Merging                    | 20%             | Randomly merges 20% of adjacent words                                     |
> | Keyboard Typos                  | 20%             | Generates typos in 20% characters using adjacent-key substitutions         |
> | Character Word Case             | 20%             | Randomly changes case for 20% of words (e.g., UPPER/lower)                |
> | Spelling Errors                 | 3 per sentence  | Introduces 3 spelling errors in each sentence                              |
> | Adverb Append                   | 1 per sentence  | Appends adverb to the end of each sentence                                |
> | Sentence Reversal               | 10% of groups   | Reverses segments using 3-word pivots in 10% of sentence groups           |
> | Sentence Repetition             | 4 sentences     | Selects and repeats 4 distinct sentences                                 |
>
> **Weakness 2** :
> Thank you for pointing out. ​L_Para​ refers to the contrastive learning loss function for paraphrase attacks, while ​L_Pert​ refers to the contrastive learning loss function for perturbation attacks. The calculation of both aligns with the two types of data generated during the "Adversarial Data Preparation" stage mentioned earlier. Together with their corresponding coefficients ​λ₁​ and ​λ₂, they constitute the MFCL loss function.
> In the original text,  we described:
> “MFCL can be divided into Paraphrase Contrastive Learning (Para-contrast) and Perturbation Contrastive Learning (Pert-contrast)”
> and
> “λ₁ and λ₂ denote the weighting coefficients for L_Para (paraphrase loss) and L_Pert (perturbation loss), respectively.”
> ​Issue:​​ The original description lacks sufficient correspondence with the "Adversarial Data Preparation" section.
> ​Corrected version​ (to be implemented if accepted for publication):
> “MFCL can be divided into Paraphrase Contrastive Learning and Perturbation Contrastive Learning, which generate two loss functions: L_Para and L_Pert, respectively.”
> and
> “λ₁ and λ₂ denote the weighting coefficients for L_Para and L_Pert, respectively.”
>
> **Weakness 3**:
> Thank you for your feedback. While leveraging contrastive learning to enhance model robustness is indeed a mainstream approach, our work introduces a core innovation: for the first time in AI-generated text detection, we propose a novel adversarial attack taxonomy fundamentally based on whether the generative source is altered. This foundational classification led to our pioneering MFCL optimization strategy—a specialized contrastive learning paradigm tailored for complex adversarial scenarios in this field. By differentially addressing distinct attack types, MFCL significantly strengthens detection robustness against intense adversarial perturbations. We contend that this deep integration of domain-specific attack mechanisms into the contrastive learning framework constitutes a substantive innovation for the domain.
>
> **Weakness 4**:
> Thank you very much for pointing this out. We have corrected both issues and performed another double-check of the entire manuscript. Should our paper be accepted, these errors will not appear in the printed version.
>
> **Question 1**:
> Thank you for your question. Based on the discriminability and stability metrics after adversarial attacks in Appendix A, Syntax Features contribute most significantly. We conducted ablation studies of statistical features on a small-scale dataset (GML130B for deepfake), with results shown below. Although improvement levels vary, all four-dimensional features enhance performance. This method primarily serves as an attempt to integrate statistical and fine-tuning approaches, and we plan to incorporate more statistical features to improve generalizability in future work.
>
> | 4-D features  | ACC   | Recall | F1   |
> |---------------|-------|--------|-------|
> | u             | 89.92 | 98.25  | 89.05 |
> | μ+σ           | 90.06 | 97.71  | 90.93 |
> | μ+σ+ ρ        | 90.43 | 98.69  | 90.70 |
> | σ+ ρ+ R       | 91.51 | 96.40  | 91.90 |
> | μ+σ+ ρ+ R     | 91.94 | 97.38  | 92.36 |
>
> **Question 2**:
> We sampled 500 HC3 points and tested OSTAR under unseen single and combined attacks. Its robustness varies by adversarial intensity:
> OSTAR's adversarially-trained features transfer well to unseen single attacks. Even against novel BERT-generated attacks (e.g., rewriting 20% sentences), it retains high accuracy. F1-score dips only ~2-3% versus known attacks, achieving 94.03—confirming stable semantic attack comprehension.
> For unseen combined attacks (perturbation→paraphrase or paraphrase→perturbation with random perturbations), statistically optimized features struggle with complex multi-step sequences. Contrastive learning remains robust, scoring F1 of 92.83 and 89.21 respectively—acceptable degradation versus seen attacks. Enhanced training coverage or advanced learning strategies could further improve performance.
> | Datasets | Original | Para   | Pert   | Bert_attack | ​Pert+Para​ | ​Para+Pert |
> |----------|----------|--------|--------|-------------|---------------|---------------|
> | HC3| 99.20| 95.89  | 97.32  | 94.03 | ​92.83​ | ​89.21 |
>
> **Question 3**:
> ​ ​ Thank you for your question. We believe directly employing large language models for machine-generated text detection is unreliable, as these models weren't designed for such tasks. Their vulnerability under attacks is substantial, as evidenced in the table below. We tested two cutting-edge large models on 1,000 DeepFake samples under paraphrasing attacks, yielding unacceptable results. Moreover, classification outcomes lack interpretability, and training such models for enhanced robustness proves challenging.
> The prompt： "You are a text classifier. Your task is to determine if the following text was generated by a machine or written by a human. "
>     "Respond with ONLY the single word 'machine' if the text is machine-generated, or 'human' if it is human-written. "
>     "Do not include any explanations, punctuation, or additional text. "
>     "Example responses:\nmachine\nhuman\n"
>
> ### Meta-Llama-3-1-8B-Instruct Classification Report
> |      Metric          | precision | recall | F1 | support |
> |---------------|-----------|--------|----------|---------|
> | machine       | 0.27      | 0.21   | 0.23     | 365     |
> | human         | 0.60      | 0.68   | 0.64     | 635     |
> | macro avg     | 0.43      | 0.44   | 0.44     | 1000    |
> | weighted avg  | 0.48      | 0.51   | 0.49     | 1000    |
>
> ### DeepSeek-R1-Distill-Qwen-7B
> | Metric        | precision | recall | F1 | support |
> |-------------|-----------|--------|-------------|---------|
> | ​machine​   | 0.33      | 0.67   | 0.45     | 365     |
> | ​human    | 0.55      | 0.23   | 0.32     | 635     |
> | macro avg     | 0.44    | 0.45   | 0.38   | 1000  |
> | weighted avg  | 0.47    | 0.39  | 0.37     | 1000    |
>
> However, utilizing large models for feature extraction remains viable. For instance, we're currently exploring their use to derive statistical features related to text evaluation metrics (commonly applied in other text processing methods) as foundations for model training.

---

> > ### Comment · Reviewer_ryiM · 2025-08-07
> > **Official Comment by Reviewer ryiM**
> >
> > Thanks for the authors' detailed clarification, which has effectively addressed the key points of concern raised earlier. I have  increased my rating.

---

### Decision · Program_Chairs · 2025-09-17

**Decision:**

Accept (poster)

**Comment:**

This paper proposes OSTAR, a framework for robust machine-generated text detection that combines statistical profiling with multi-faceted contrastive learning. The motivation is clear, the methodology is technically sound, and experiments on multiple datasets and attack scenarios are comprehensive.

Reviewers raised concerns about clarity of intrinsic feature claims, limited attack diversity, and computational overhead, but the rebuttal addressed these points with additional analyses and implementation details.

Overall, the paper is well-motivated, clearly written, and demonstrates strong empirical performance. I recommend acceptance.